# Minimal Images in Deep Neural Networks: Fragile Object Recognition in Natural Images

**Sanjana Srivastava**[1,3], **Guy Ben-Yosef**[1,2][*] **Xavier Boix**[1,3,4*]
[1]Center for Brains, Minds, and Machines (CBMM)
[2]Computer Science and Artificial Intelligence Laboratory, Massachusetts Institute of Technology, USA
[3]Department of Brain and Cognitive Sciences, Massachusetts Institute of Technology, USA
[4]Children's Hospital, Harvard Medical School, USA
`{sanjanas, gby, xboix}@mit.edu`

## Abstract

The human ability to recognize objects is impaired when the object is not shown in full. "Minimal images" are the smallest regions of an image that remain recognizable for humans. Ullman et al. (2016) show that a slight modification of the location and size of the visible region of the minimal image produces a sharp drop in human recognition accuracy. In this paper, we demonstrate that such drops in accuracy due to changes of the visible region are a common phenomenon between humans and existing state-of-the-art deep neural networks (DNNs), and are much more prominent in DNNs. We found many cases where DNNs classified one region correctly and the other incorrectly, though they only differed by one row or column of pixels, and were often bigger than the average human minimal image size. We show that this phenomenon is independent from previous works that have reported lack of invariance to minor modifications in object location in DNNs. Our results thus reveal a new failure mode of DNNs that also affects humans to a much lesser degree. They expose how fragile DNN recognition ability is for natural images even without adversarial patterns being introduced. Bringing the robustness of DNNs in natural images to the human level remains an open challenge for the community.

## 1 Introduction

Deep neural networks (DNNs) have reached tremendous success in recognizing and localizing objects in images. The fundamental approach that led to DNNs consists of building artificial systems based on the brain and human vision. Yet in many important aspects, the capabilities of DNNs are inferior to those of human vision. A promising strand of research is to investigate the similarities and differences, and by bridging the gaps, further improve DNNs (Hassabis et al., 2017).

Studying cognitive biases and optical illusions is particularly revealing of the function of a visual system, whether natural or artificial. Ullman et al. (2016) present such a striking phenomenon of human vision, called "minimal images" (Ullman et al., 2016; Ben-Yosef et al., 2018; Ben-Yosef & Ullman, 2018). Minimal images are small regions of an image (*e.g.* 10x10 pixels) in which only a part of an object is observed, and a slight adjustment of the visible area produces a sharp drop in human recognition accuracy. Figure 1 provides examples of human minimal images.

Ullman et al. (2016) show that DNNs are unable to recognize human minimal images, and the DNN drop in accuracy for these minimal images is gradual rather than sharp. This begs the question of whether the sharp drop in accuracy for minimal images is a phenomenon exclusive to human vision, or there exist distinct but analogous images that produce a sharp drop in DNN accuracy.

In this paper, we provide evidence for the latter hypothesis by showing that there is a large set of analogs to minimal images that affect DNNs. These are different from the minimal images for humans in several aspects, namely region size and location, and frequency and sharpness of the drop in accuracy. We find that a slight adjustment of a one-pixel shift or two-pixel shrink of the visible

---

[*]Equal Contribution

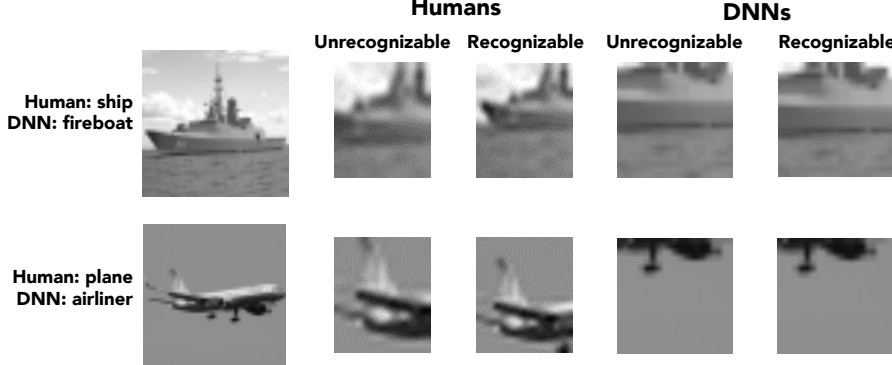

Figure 1: *Qualitative examples of human and DNN minimal images.* Human minimal images and their less-recognizable sub-images from Ullman et al. (2016); and DNN analogs for the same images.

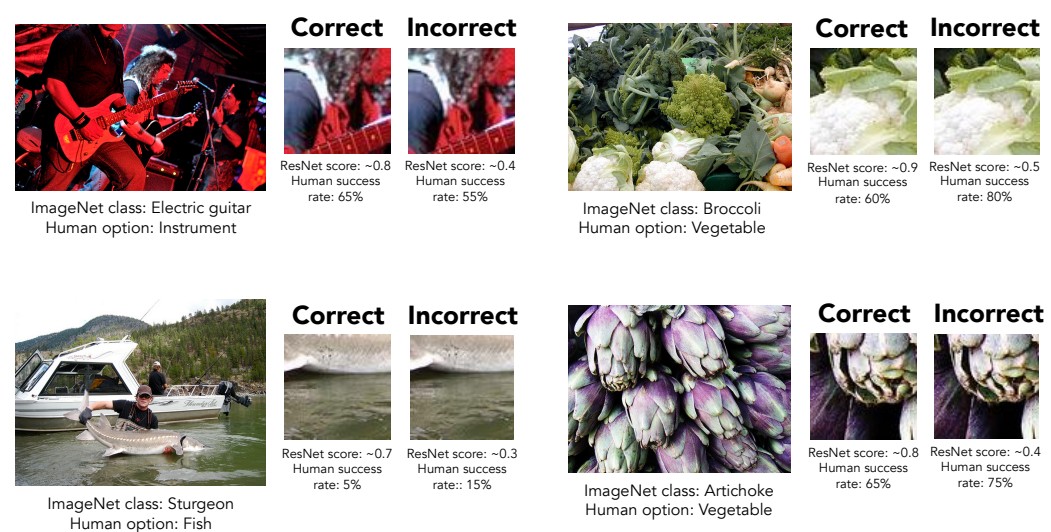

Figure 2: *Qualitative examples of fragile recognition images (FRI).* Examples are for ResNet (He et al., 2016) analogs to human minimal images. The incorrectly classified image region is slightly (two pixels) smaller than the correctly classified crop. Even when DNNs show a significant change in confidence between regions, humans recognize them at similar rates.

image region produces a drop in DNN recognition accuracy in many image regions. Figure 2 shows several examples of minimal image analogs of state-of-the-art DNNs.

The adjustments of the visible area that affect DNNs are almost indistinguishable to humans and can occur in larger regions than the "human minimal" region. To describe this phenomenon we introduce *fragile recognition images (FRIs)* for DNNs, which are more general than minimal images:

**Fragile Recognition Image (FRI):** *A fragile recognition image is a region of an image for which a slight change of the region's size or location in the image produces a large change in DNN recognition output.*

This definition is more general than the definition of minimal images for humans, as the latter is included in the definition for DNNs. Minimal images are the case of fragile recognition in which the slight change is a reduction in the size of the region, and the minimal image is one of the smallest possible FRIs. In human vision, the more general definition of fragile recognition that we are introducing here is not useful because human minimal images appear only when the visible area of the object is small.

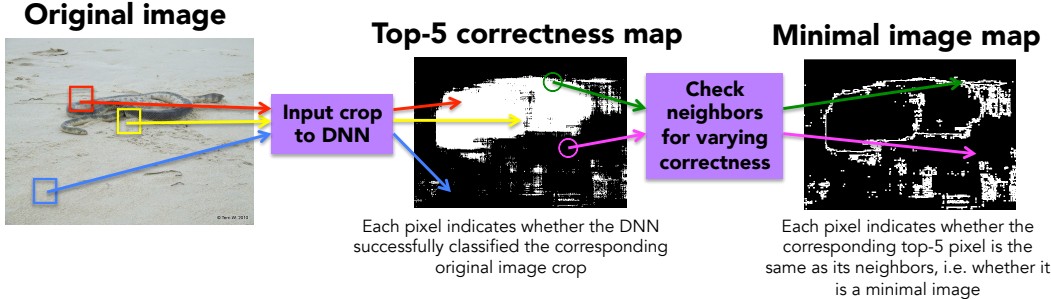

Figure 3: *The process of generating a fragile recognition image (FRI) map from a full image.* The FRI map shows FRIs for which a one-pixel shift in any direction of the visible region produces an incorrect classification (*i.e.* loose shift - see definition in 2). The white pixels indicate FRIs and black pixels indicate non FRIs.

Since FRIs are hardly distinguishable to humans but cause DNNs to fail, there is a connection with so-called *adversarial examples* presented by Szegedy et al. (2013). Adversarial examples are images with small synthetic perturbations that are imperceptible to humans, but produce a sharp drop in DNN recognition accuracy. There are several types, *eg*. (Goodfellow et al., 2014; Liu et al., 2016), and the strategies to alleviate them are not able to fully protect DNNs (Tramèr et al., 2017; Madry et al., 2017). Furthermore, humans may suffer from adversarial attacks; human recognition ability is shown to be impaired by perturbations under rapid image presentation (Elsayed et al., 2018). Unlike these adversarial examples, fragile recognition arise in natural images without introducing synthetic perturbation. This causes new concerns for use of DNNs in computer vision applications.

We evaluate FRIs in ImageNet (Deng et al., 2009) for state-of-the-art DNNs, specifically VGG-16 (Simonyan & Zisserman, 2015), Inception (Szegedy et al., 2014), and ResNet (He et al., 2016). Results show that FRIs are abundant and can occur for any region size. Furthermore, we investigate whether fragile recognition is related to the lack of invariance to small changes in the object location, which has been recently reported in the literature to affect DNNs (Engstrom et al., 2017; Azulay & Weiss, 2018). Our results demonstrate that fragile recognition is independent from this phenomenon: bigger pooling regions reduce most of the lack of invariance to changes in object location, while pooling only marginally reduces fragile recognition. We also show that known strategies to increase network generalization, *i.e.* adding regularization and data augmentation, reduce the number of FRIs but still leave far more than humans have. These results highlight how much more fragile current DNN recognition ability is than human vision.

## 2 METHODS: EXTRACTING FRAGILE RECOGNITION IMAGES

In this Section we introduce the method for extracting FRIs for DNNs. The method for extracting human minimal images employs a tree search strategy (Ullman et al., 2016). The full object is presented to human subjects and they are asked to recognize it. If at least $50\%$ of subjects recognize it correctly, smaller crops of the object, called descendants, are tested. If any of the smaller region is recognizable, the crop size is further reduced and tested. Once an image crop is found such that it is recognizable and all of its descendants are not recognizable, it is considered a human minimal image.

Our FRI extraction method for DNNs relies on an exhaustive grid search. This is possible in DNNs due to parallelization across multiple GPUs, and would be prohibitively time-consuming to replicate with humans. The grid search consists on a two-step process: first, every possible square region is classified by the DNN and the correctness is annotated in the *correctness map*. From the correctness map, each region's correctness is compared with the region's slightly changed location or size in order to determine if there has been a change of the correctness. FRIs are regions that are classified correctly and a small change causes failure, as well as regions that are classified incorrectly and a small change causes success.

The two step process to detect FRIs is summarized in Figure 3. We now detail the two steps:

**1. Correctness map generation.** An exhaustive grid search is performed to see if each possible square image region of a fixed size is classified correctly by a given DNN. After extracting the

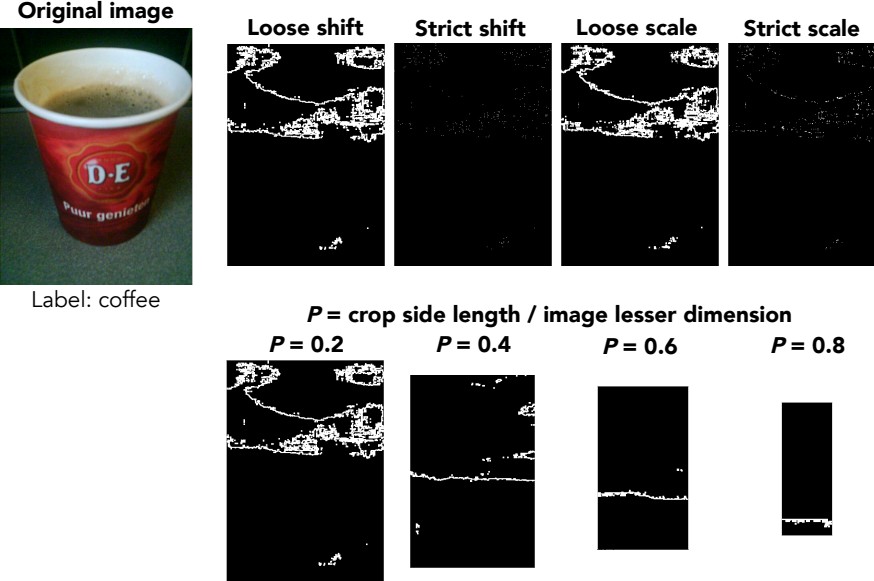

Figure 4: *Fragile recognition image (FRI) maps*. White pixels indicate FRIs and black pixels indicate non FRIs. First row: visualizations of all four types of FRI maps with $P = 0.2$. Second row: visualizations of loose shift FRI maps for $P = 0.2, 0.4, 0.6, 0.8$. Maps generated with Inception (Szegedy et al., 2014).

region from the image, the region is resized to be of the size required by the network. The region size is parametrized by $P$, which is defined as the proportion of the image occupied by the region, *i.e.* $P = S/\min(h, w)$ where $h$ and $w$ are the height and width of the input image, and $S$ is the region's side length. The results are arranged in a map such that a given map pixel contains the binary correctness for the square region centered at the corresponding pixel in the original image. The resulting map is of dimension $(h - S) \times (w - S)$ due to padding loss. The first two panels of Figure 3 show a visualization of the correctness map generation process.

**2. Fragile recognition image (FRI) extraction from correctness maps.** We define different variations of FRIs, depending if they are based on changes on the location or size of the image region, and on how strict we are when evaluating the changes in the correctness of DNN:

–*Shift or Shrink*: we define two types of "small changes" of the region that affect DNN correctness. "Shift" is a one-pixel translation of the region location; "shrink" is a two-pixel reduction of the region side length within the region's original boundaries, in a fixed full image size.

–*Strict or Loose*: the visible region can be shifted in various directions, or shrunk in different ways while remaining within its original boundaries. "Loose" FRIs are regions such that there exists a small change that flip network correctness. "Strict" FRIs are regions such that network correctness is flipped for all small changes.

These definitions yield four fragile recognition types: loose shift, loose shrink, strict shift, and strict shrink. Note that strict shrink is the most analogous to human minimal images. The correctness maps are used to detect fragile recognition due to shifts by comparing neighbouring pixels in a correctness map, and due to shrinks by comparing correctness maps at two slightly different region sizes. We use *fragile recognition image (FRI) maps* to visually represent the result of the grid search that extracts FRIs. As shown in the FRI map of Figure 3, each pixel of the map indicates whether the corresponding window in the original image is an FRI. The second and third panel show a visualization of the FRI map generation process. Figure 4 shows an ImageNet image and each of its FRI maps.

## 3 FRAGILE RECOGNITION IMAGES OF STATE-OF-THE-ART DNNS

In this Section, we will first discuss occurrence of fragile recognition for state-of-the-art object recognition models through results based on ImageNet (Deng et al., 2009). Then, we analyze if data

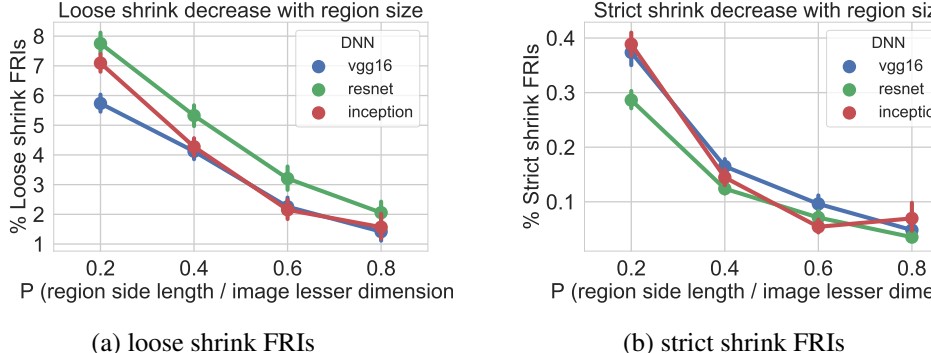

(a) loose shrink FRIs                                    (b) strict shrink FRIs

Figure 5: *Fragile recognition images (FRIs) in ImageNet.* FRIs are generally less frequent for larger regions of the image. (a) Loose shrink FRIs, which indicate the general fragility of DNNs to a slight reduction in the visible region. (b) Strict shrink FRIs, which are the equivalent to human minimal images. Shift FRIs also follow this pattern (see Figure A.1).

augmentation and regularization help reduce fragile recognition through experiments in CIFAR-10 Krizhevsky (2009).

## 3.1 FRAGILE RECOGNITION IMAGES FOR STATE-OF-THE-ART DNNs IN IMAGENET

The following experiments are performed on 500 images, randomly sampled from ImageNet's validation set, which consists of 50,000 images and ground-truth object bound-in boxes (Deng et al., 2009). The images are sampled from 10 supercategories (dog, snake, monkey, fish, vegetable, musical instrument, boat, land vehicle, drinks, furniture), each of which covers a set of ImageNet categories. Correctness is measured as top-5 accuracy, which is commonly used in ImageNet. FRIs are extracted for three architectures: VGG-16 (Simonyan & Zisserman, 2015), Inception (Szegedy et al., 2014), and ResNet (He et al., 2016). Experiments are run in eight K80 NVIDIA GPUs. The exhaustive grid search takes about 5 minutes for one image using all eight GPUs. In the following paragraphs, we report the results of the experiments.

We evaluate how much a DNN is affected by fragile recognition by quantifying the proportion of possible image regions that are FRIs, *i.e.* we quantify the amount of regions affected by a shift or shrink. Recall that we consider a region to be an FRI when the classification changes from correct to incorrect, and also from incorrect to correct. As expected, we found that both of these cases are approximately equally probable under all tested conditions.

In Figure 5, we show the percentage of shrink FRIs in the image for a given region size, $P$; in Figure A.1 we show the same for shift FRIs. All networks are significantly affected by FRIs, with ResNet being the most: almost one out of 50 regions of a size that covers almost the entire image ($P = 80\%$) can affect ResNet. When the size of the image region is $P = 20\%$, FRIs are even more frequent (8 times more), *i.e.* almost two out of 25 regions in the image is a loose FRI for ResNet.

Comparing Figure 5a and Figure 5b, we see that the proportion of strict FRIs is less than that of loose FRIs because of the more stringent definition. The results show that there are many regions in an image for which the network is very sensitive to slight changes in the region, *eg.* one out of 250 image regions of size $P = 20\%$ will be misclassified by ResNet when the region is slightly shrunk. Recall that strict shrink FRIs are the equivalent case of minimal images in humans. Thus, these results demonstrate the existence of many minimal images analogs in DNNs.

Note that smaller FRIs (lower $P$) are much more frequent than larger ones, as observed across the different network architectures and types of FRIs. This trend is expected: one-pixel shifts and two-pixel shrinks are proportionally larger changes for smaller regions, and larger regions generally allow for more high-level features to be included.

We verified that FRIs are not an artifact of the algorithm that resizes the region to the size required by the DNN ($224 \times 224$ pixels for VGG-16). We took regions of side length 224 and removed any resizing before input, and we observed that this procedure produces the same results we reported.

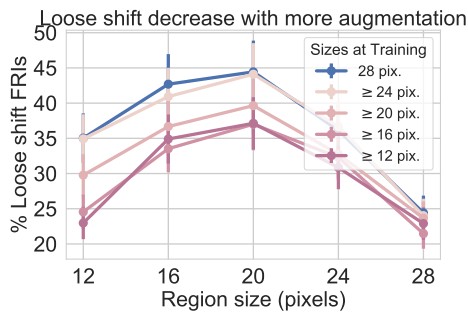 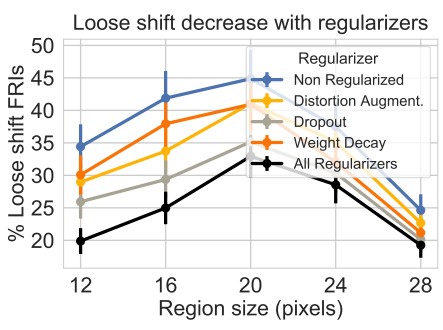

(a) data augmentation by cropping image regions      (b) DNN with regularization

Figure 6: *Impact of data augmentation and regularization in fragile recognition (CIFAR-10).* FRIs can be mitigated but not eliminated. (a) shows that augmenting the training set with crops of FRI sizes reduces overall FRI occurrence, but many FRIs remain. (b) shows that commonly used regularizers may also reduce the percentage of FRIs, but they still occur.

Besides the qualitative examples in Figure 1 and Figure 2, we display a more varied set of qualitative examples in Figure A.10. We observe that FRIs are usually located within object boundaries but can also be found in the background. This is because DNNs are able to recognize regions that only contain background (Zhu et al., 2016), as they have been shown to exploit dataset biases.

In Figure A.9 we display the DNN's output confidence in the true class for individual regions in map form. We see that sharp drops in confidence are frequent within an image, providing a basis for minimal images. Finally, in Figure A.11 we show qualitative examples of the activation maps at different layers of the DNN of the correctly classified crop and its shifted version. These examples show what we have observed in all cases: the activation maps are imperceptibly similar at the first layers but are clearly different at the last layers.

Finally, we show qualitative examples to illustrate that FRIs are a concern for computer vision systems based on similar DNNs architectures. We conducted a test on detection algorithms, to validate that FRIs dramatically affect both the location and the label of the detected objects. In Figure A.12 we show FRIs for the widely used object detector called "YOLO" (Redmon & Farhadi, 2017).

## 3.2 FRAGILE RECOGNITION WITH DATA AUGMENTATION AND REGULARIZATION

We now investigate if data augmentation and regularization help alleviate fragile recognition. In this experiment we use the CIFAR-10 dataset (Krizhevsky, 2009), which contains 10 object categories, $50,000$ training images and $10,000$ testing images of size $32 \times 32$ pixels. The evaluation criteria is top-1 accuracy. We use CIFAR-10 for convenience and without loss of generality, since the data augmentation and regularization we test are used in the models tested in ImageNet.

We reproduce the AlexNet version for CIFAR-10 introduced by Zhang et al. (2016), which consists of two convolutional-pooling-normalization layers followed by two fully connected layers. In the following, all regularizers and data augmentation are turned off unless stated otherwise. We will analyze the impact of each to fragile recognition.

For data augmentation, we augment the training dataset with FRI regions of at least a given size. For regularizers, we add weight decay, dropout, and distortions (e.g. reflections, altered brightness, altered contrast). Both the data augmentation and the regularizers improve the accuracy of the DNN for the different crop sizes (Figure A.2). Figure 6 shows results for loose shift FRIs (see Figure A.3 for shrink FRIs); both data augmentation and regularization have a clear impact on FRI occurrence in all cases. The largest general improvement comes for the 12-pixel region size, as data augmentation and regularization both lead to decrease of more than 12% of FRIs. For regularizers specifically, dropout provides the most individual improvement. However, all of these generalization efforts still allow a high failure rate. They are also unable to bring the FRI effect to the human level, as the DNN FRIs are indistinguishable for humans. This is further discussed in Section 5.

In ImageNet, smaller region sizes lead to higher FRI occurrence. In CIFAR-10, FRI occurrence in very small regions (smaller than 20 pixels) decreases with size. This is because 12- and 16-pixel

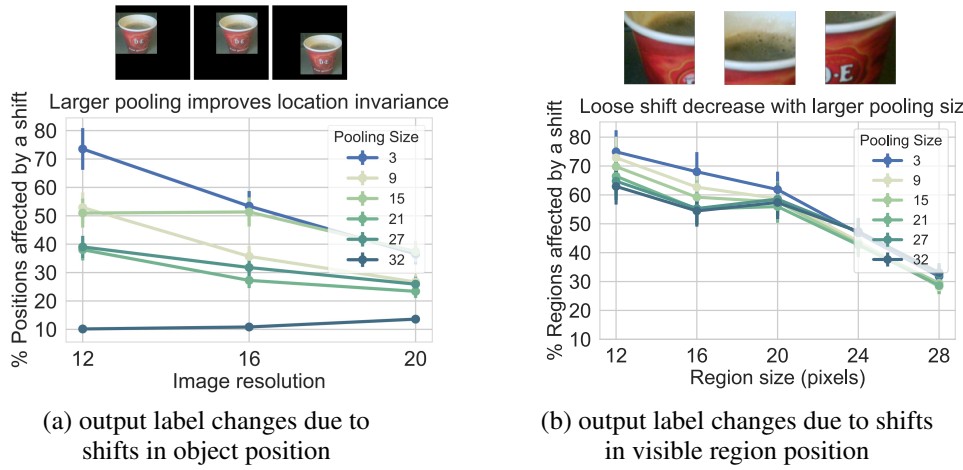

(a) output label changes due to
shifts in object position

(b) output label changes due to shifts
in visible region position

Figure 7: *Comparison of location invariance and fragile recognition images (CIFAR-10).* FRIs are distinct from translation-based adversarial examples. (a) shows that maximum pooling makes the DNN almost entirely robust to translation by maximizing location invariance; the remaining $10\%$ of translations that do cause variation are generally at the edge of the black background and are due to edge effects (see Figure A.7). By contrast, (b) shows that larger pooling sizes provide little to no robustness to slight shifts of the visible region.

regions are prohibitively small, so the number of correctly classified regions is severely reduced; consequently, there are fewer opportunities for FRIs to occur at all.

## 4  FRAGILE RECOGNITION IS NOT LACK OF OBJECT LOCATION INVARIANCE

We now focus on the relationship of fragile recognition with recent works that show that DNNs are affected by small changes of the object location, scale and orientation (Engstrom et al., 2017; Azulay & Weiss, 2018). The phenomenon described in previous works is referred to as lack of invariance due to affine transformations of the object. Here, we focus only on location changes, as it will allow to distinguish FRIs and these previous works.

The procedure to evaluate location invariance is the following, quoting from Azulay & Weiss (2018): "we embed the original image in a larger image and shift it in the image plane (while filling in the rest of the image with a simple inpainting procedure)". The embedding of the image can also be done with a black empty background as in (Engstrom et al., 2017), or using videos in which the background is static and only the object is moving (Azulay & Weiss, 2018). In fragile recognition, both the object and the background, *i.e.* the entire image, change due to the shift or shrink of the image region. In previous works, only the object changes location. We introduce an experiment that reveals that this subtle difference makes fragile recognition an independent and more complex phenomenon than lack of location invariance.

**Pooling induces location invariance.** DNN architectures with large pooling regions are known to induce invariance to the object location, *cf.* (Anselmi et al., 2015). A pooling layer operates independently for each feature by extracting the maximum activation across the spatial dimensions. For a pooling region of the same size as the full image, the network response is invariant to the object position within the image frame. This is under the assumption that there is enough spatial resolution in the responses in order to guarantee that the responses after the shift do not vary except for the shift. See (Azulay & Weiss, 2018) for a detailed explanation. Note that this assumption is not fulfilled in FRIs, as a shift of the image region introduces and removes patterns in the borders of the image, which can produce changes in the responses after pooling. We now show this experimentally.

**Experimental results.** We evaluate the network trained on CIFAR-10 and introduced in the previous section with different pooling region sizes for the second pooling layer. These sizes range from three pixels to the entire image; theoretically, this should produce full location invariance. In order to guarantee sufficient spatial resolution to obtain location invariance, we use a stride of one pixel for the convolutional layers and add padding. Adding larger pooling regions does not reduce network accuracy (see Figure A.5a), but as we show next, it massively reduces the lack of location invariance.

We evaluate the DNN by embedding the object in a black background at all possible locations, and we quantify for how many locations a one-pixel shift of the object location produces a change in the output of the DNN (similar to the method in Section 2). In this experiment we evaluate changes to the output label and not correctness, in order to accurately match the definition of invariance.

Figure 7a shows that as expected, the lack of invariance is dramatically reduced when using large pooling regions. For embedded 12-pixel images, $75\%$ of the one-pixel shifts affect the network with small pooling regions, while only $10\%$ of the one-pixel shifts affect the output of the network. Pooling is not completely location invariant due to boundary effects near the perimeter of the image. We show qualitative examples of this reduction in Figure A.7.

In Figure 7b, we see that increasing the pooling size only slightly decreases FRIs. The region size with a larger decrease is 12 pixels (note that the embedding size can not be directly compared with the region size of FRIs). For this region size, there are approximately $10\%$ less FRIs, but a significant amount still remain (about $65\%$ of the regions are FRIs). Since the pooling mechanisms that largely reduced the lack of invariance are not effective for fragile recognition, we can conclude that FRIs are a more complex phenomenon than lack of location invariance.

Note that for small region sizes the amount of FRIs increases, which is different from what we observe in the data augmentation and regularization experiment in the previous section. This is because in Figure 7b we report change in the output label rather than change in the correctness as in previous experiments. See Figure A.5b for the effect of pooling on FRI occurrence, which is in accordance to previous results.

Finally, we control that the differences we observe between lack of invariance and FRIs are not caused by use of zero-padding in one case and not the other. In Figure A.6, we evaluate FRIs with zero-padding instead of up-scaling the cropped region. The results support the same conclusions as for FRIs with up-scaling.

## 5 COMPARING FRAGILE RECOGNITION IN HUMANS AND DNNS

In this section, we further compare the FRIs found for DNNs with the human minimal images found by Ullman et al. (2016). Recall that minimal images are equivalent to strict shrink FRIs of small size. Both humans and DNNs are susceptible to small image changes, but these two sets of fragile images are not necessarily the same in humans and in DNNs. As shown by Ullman et al. (2016); Ben-Yosef et al. (2018), when DNNs are trained and tested on human minimal images, *i.e.* images that humans can still recognize, DNNs are unable to recognize the objects.

Here we show the converse, namely that when humans are tested on DNN fragile recognition images, they do not exhibit the same fragile response. This can be seen in our qualitative examples: DNN fragile recognition images and their incorrectly-classified counterparts are difficult to distinguish. To verify this, we randomly sample 40 DNN shrink FRIs generated from ImageNet. We present the correctly-classified FRI and the slightly changed image (both resized to $100 \times 100$ pixels) to separate groups of 20 subjects in Mechanical Turk (Buhrmester et al., 2011). The subjects annotate the images using one of our 10 supercategory labels. The results show a small gap in correct recognition: on average, success rates among humans for DNN fragile recognition images and incorrect counterparts differ by $14.5\%$. DNNs experience an average gap in confidence of $56\%$ for the same pairs.

Furthermore, we directly compare human and DNN minimal images of the same image. We use six of the original images from Ullman et al. (2016) that were used to find human minimal images. We identify the DNN FRIs for these. Comparing DNN FRIs and human minimal images of the same region size, $P$, we find two additional differences: first, DNNs have more FRIs (5.3 minimal images per image for humans on average, 13.4 for DNNs). Second, DNN FRIs differ in location within the object region (Intersection Over Union between human minimal image maps and DNN FRI maps was small, $< 6\%$ for all tested images). For example, while human minimal images are centered in meaningful object parts (e.g. the eagle head and wing in Figure A.8 row 1, column 1), DNN FRIs contain mostly a background set of pixels (see Figure A.8 row 1, column 2).

Finally, we control that current DNNs architectures that are based on human vision are also severely affected by FRIs. This is shown in Figure A.4 for the recently introduced multi-scale eccentricity dependent DNN that mimics these aspects of human vision (Chen et al., 2017).

# 6 CONCLUSIONS

We have demonstrated that both humans and DNNs are affected by slight changes of the visible region. Since the human visual system also exhibits similar fragile recognition behavior, this might be expected or even desired. Our results have revealed that the fragile recognition level in humans is fundamentally different from the one in DNNs in terms of size, position and frequency. Furthermore, data augmentation, regularization and larger-size pooling regions alleviate fragile recongition in DNNs, but are not sufficient to close the gap with humans. Thus, making DNNs robust like humans remains a challenge for the community. Finally, we have shown that fragile recognition is a more complex phenomenon than object location invariance, which exposes new concerns of the recognition ability of DNNs in natural images, even without adversarial patterns being introduced.

ACKNOWLEDGMENTS

We are grateful to Tomaso Poggio and Shimon Ullman for helpful feedback and discussions. This work is supported by the National Science Foundation Science and Technology Center Award CCF-123121, the MIT-IBM Brain-Inspired Multimedia Comprehension project, and the MIT-Sensetime Alliance on Artificial Intelligence.

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

## A    APPENDIX

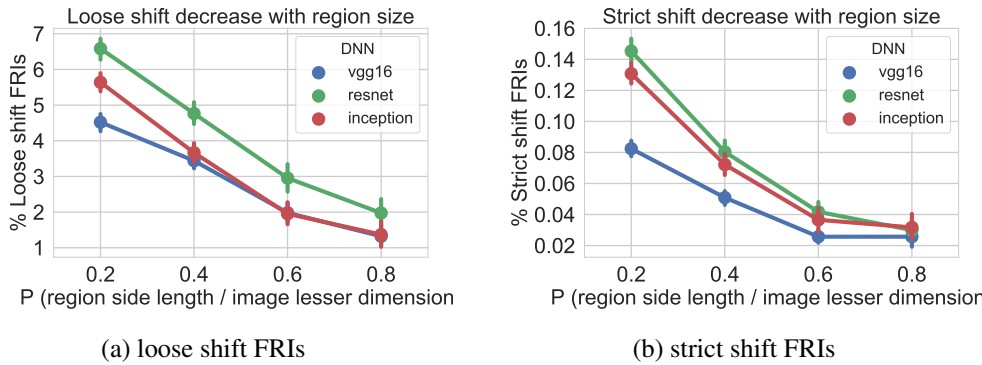

(a) loose shift FRIs                    (b) strict shift FRIs

Figure A.1: *Fragile recognition images (FRIs) in ImageNet.* All types of FRIs decrease with increasing crop size. (a) Loose shift FRIs, indicating the general fragility of DNNs to a slight shift of the visible region. (b) Strict shift FRI.

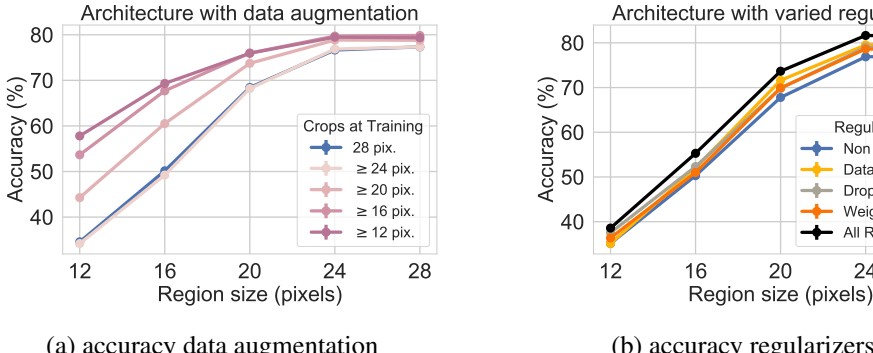

(a) accuracy data augmentation

(b) accuracy regularizers

Figure A.2: *Accuracy for data augmentation and regulariers (CIFAR-10).* Data augmentation using smaller crops and the regularizers help improving the accuracy at different region sizes.

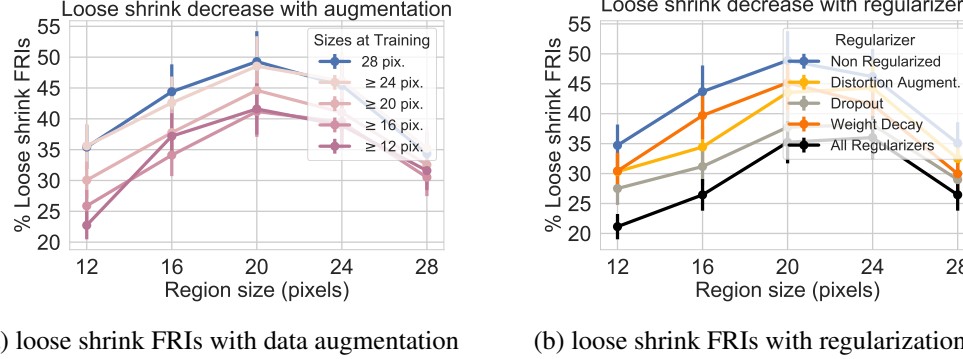

(a) loose shrink FRIs with data augmentation

(b) loose shrink FRIs with regularization

Figure A.3: *Impact of the data augmentation and regularization in fragile recognition (CIFAR-10).* As with loose shift FRIs, data augmentation using smaller crops and standard regularizers helps reduce loose shrink FRIs, but does not eliminate the problem by any means.

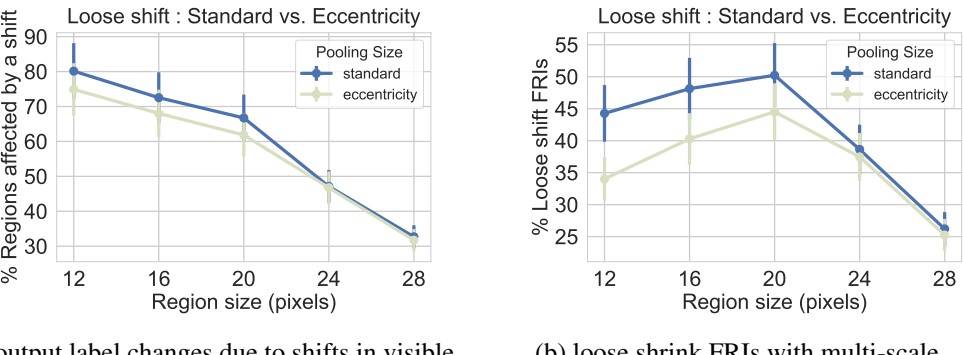

(a) output label changes due to shifts in visible region position for multi-scale eccentricity

(b) loose shrink FRIs with multi-scale eccentricity

Figure A.4: *Impact of multi-scale eccentricity dependent architecture (CIFAR-10).* DNN architectures inspired by human vision are eccentricity dependent (*i.e.* high image resolution in the center and low image resolution in the periphery) and process the image at multiple scales (Chen et al., 2017). We use the architecture with two scales, one that covers a $20 \times 20$ pixels at full resolution, and the other scale covers the entire image at half resolution. The two scales were combined by the first convolutional layer. This architecture helps reducing FRIs, but there exist a big gap between this architectures and humans in terms of FRIs.

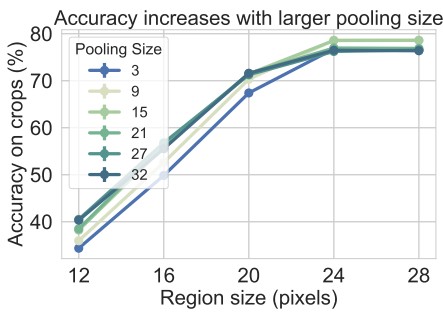

(a) accuracy of DNN on small crops

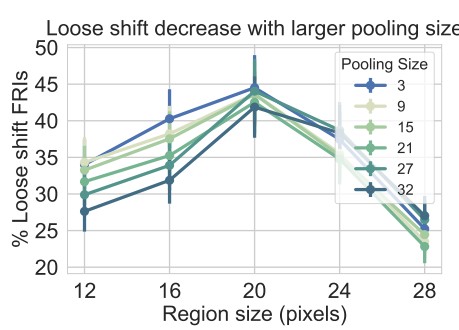

(b) changes in correctness due to small shifts

Figure A.5: *FRIs with DNN with large pooling regions.* Pooling has some mitigating effect on FRIs for lower crops, but does not significantly reduce them. (a) shows that pooling size also has little effect on DNN accuracy on crops. (b) shows that a larger pooling size provides little to no reduction of FRIs depending on crop size.

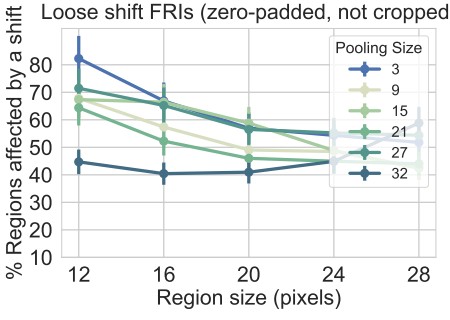

(a) output label changes due to
shifts with **zero-padding**

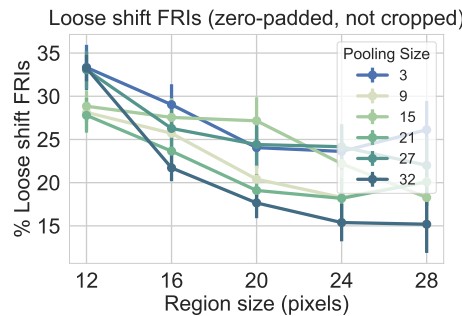

(b) changes in correctness due to small shifts
with **zero-padding**

Figure A.6: *FRIs with zero-padding for DNN with large pooling regions.* FRIs with zero-padding have similar behaviour as FRIs with up-scaling. (a) Pooling has some mitigating effect on FRIs with zero-padding for lower crops, but does not significantly reduce them. (b) shows that a larger pooling size provides little to no reduction of FRIs depending on crop size.

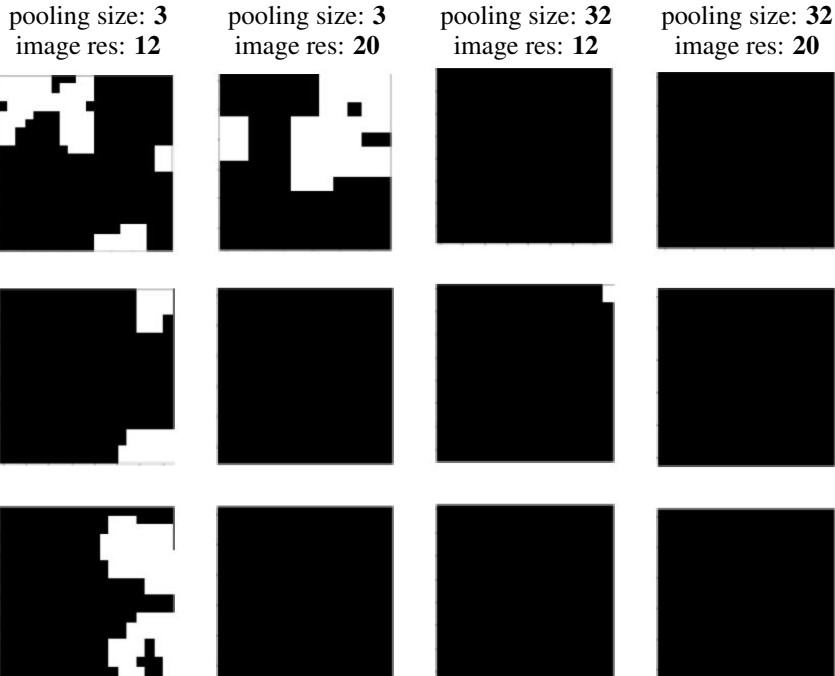

Figure A.7: *Effect of pooling to location invariance.* Pooling almost eliminates output label changes due to translation. The above maps are equivalent to FRI maps, except each pixel corresponds to a downscaled image with a given side length centered at that pixel, rather than a crop centered at that pixel. The remaining variation seen in 7b (around $10\%$ of positions cause change in output label even with the largest pooling region) appear to be due to edge effects, as seen in row 2 and column 3 of the above table.

**DNN strict shrink FRI map**   **Human minimal image map**

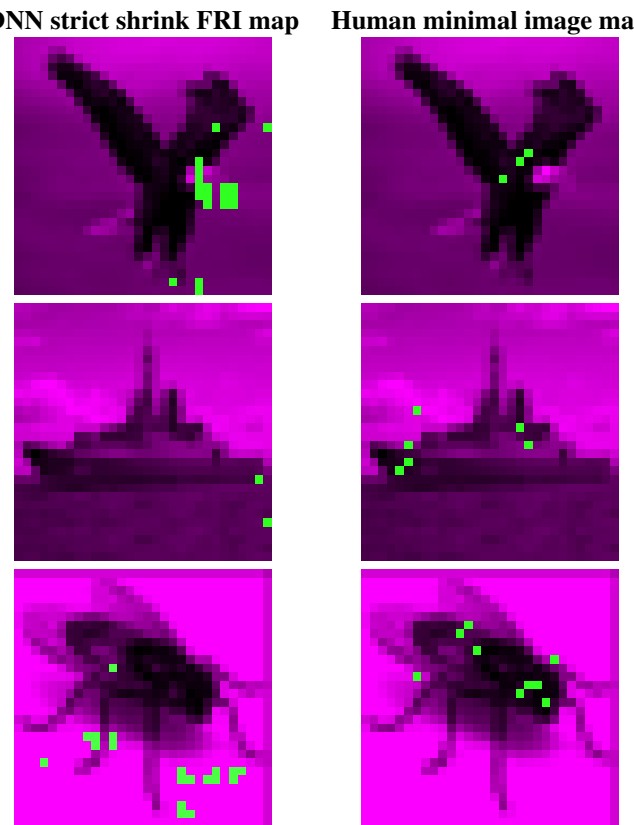

Figure A.8: *Comparison of DNN and human minimal image maps.* Human minimal images and the equivalent to DNNs (strict shrink FRIs) differ in frequency and location. The left column shows DNN strict shrink FRI locations in green; the right column shows human minimal image locations in green. Both show non-minimal image locations in pink. In these examples, we see how human minimal images tend to occur on meaningful features of the object, whereas DNN strict shrink FRIs occur in locations that do not seem to be intuitive for humans. Note that the human minimal images shown have $0.35 \leq P \leq 0.45$, and the DNN FRIs shown have $P = 0.4$. More minimal images/FRIs exist for both perceptual systems at other scales.

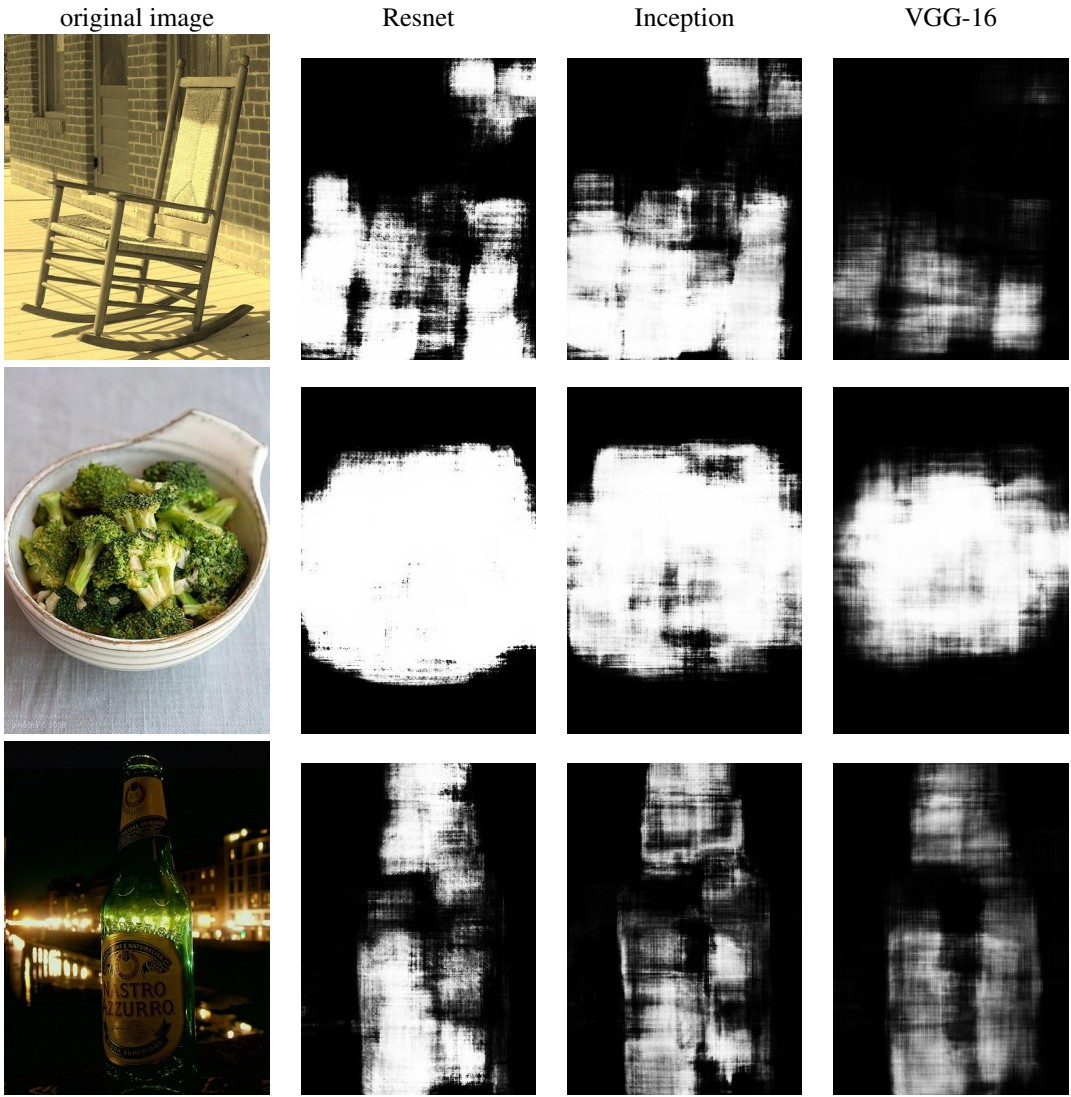

Figure A.9: *Confidence score of the DNNs for different regions.* Sharp drops of the confidence score between image regions that are one or two pixels apart occur frequently. We introduce a map analogous to the correctness map for the confidence score. Each pixel in the map indicates the output score of the DNN for the ground-truth class after the softmax layer, for the image region centered at the corresponding pixel. The figure shows several of these maps given a region size of $P = 0.2$ (the same conclusions are extracted for any of the other values of $P$ we test in the paper). These maps clearly show sharp drops of the confidence score.

Loose shift FRIs; $P = 0.2$, ResNet

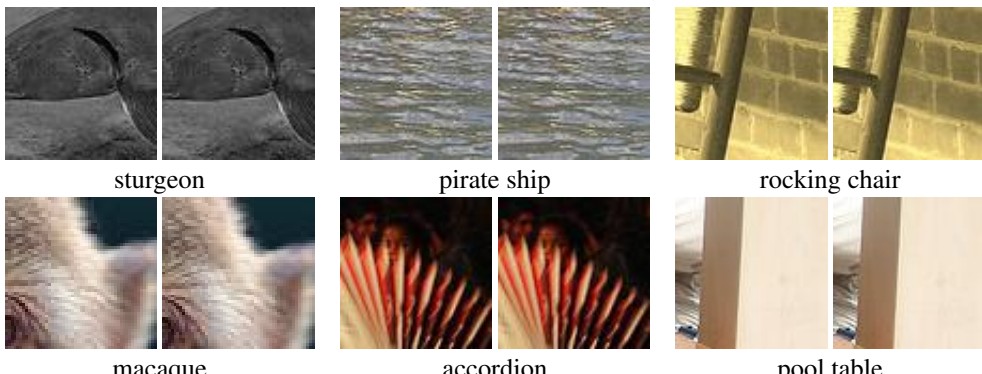

sturgeon  pirate ship  rocking chair

macaque  accordion  pool table

Loose shift FRIs; $P = 0.6$, ResNet

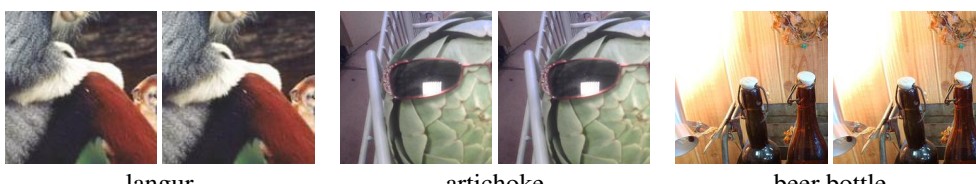

langur  artichoke  beer bottle

Loose shrink FRIs; $P = 0.2$, ResNet

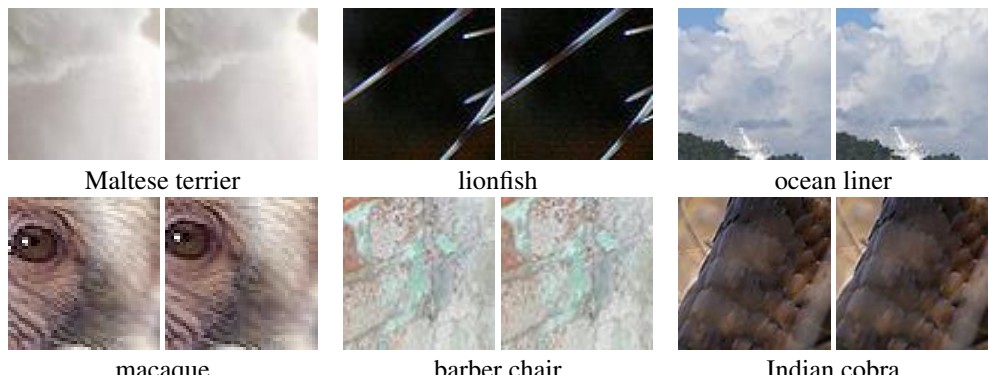

Maltese terrier  lionfish  ocean liner

macaque  barber chair  Indian cobra

Loose shrink FRIs; $P = 0.6$, Resnet

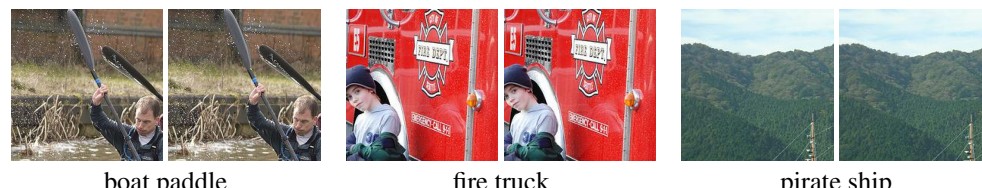

boat paddle  fire truck  pirate ship

Loose shrink FRIs; $P = 0.2$, VGG-16

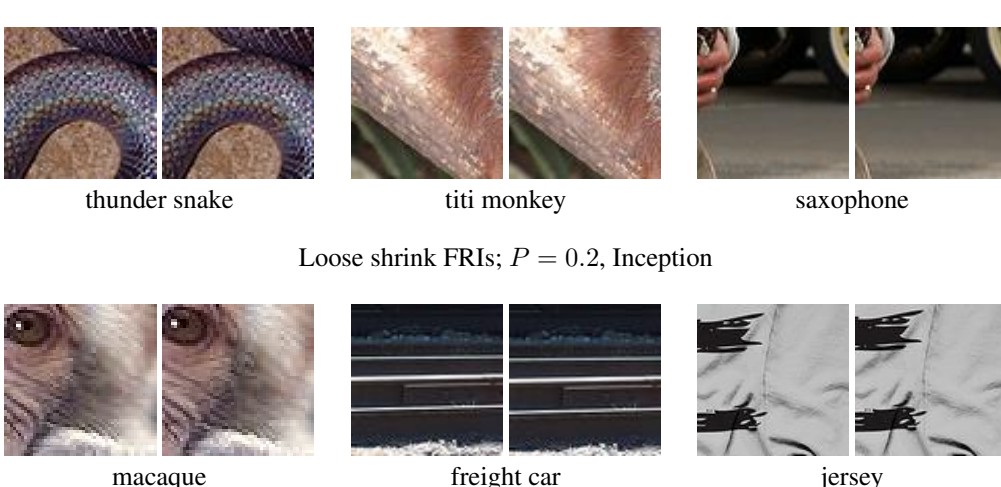

Loose shrink FRIs; $P = 0.2$, Inception

Figure A.10: *Examples of FRIs.* In each pair, the left image is correctly classified and the right image is incorrectly classified. The true class is provided below the crops. We see that for smaller $P$ (0.2), FRIs often occur near the edges of the object and contain some fraction of it. However, for larger $P$ (0.6) which generally occur at a lesser rate, the FRIs contain more of the object. They tend to contain multiple objects and/or confounding objects (*eg.* people) that the DNN is not trained on.

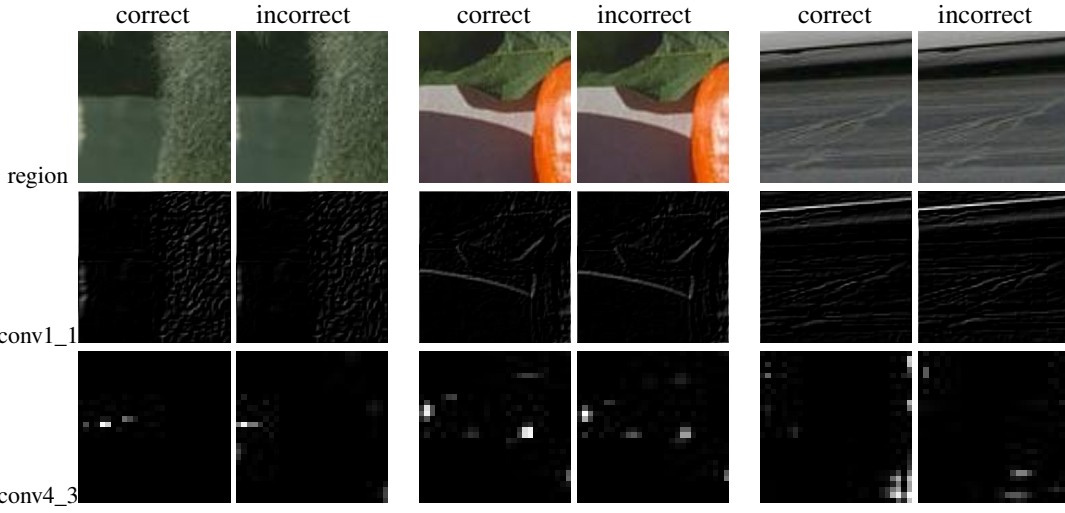

Figure A.11: *Activations for loose shift FRIs and their incorrect counterparts.* The FRIs were generated using VGG-16 with $P = 0.2$. The first row shows the region themselves, the second row shows the activations from the first convolutional layer, and the third row shows the output from the tenth convolutional layer.

change in classification score        change in bounding box

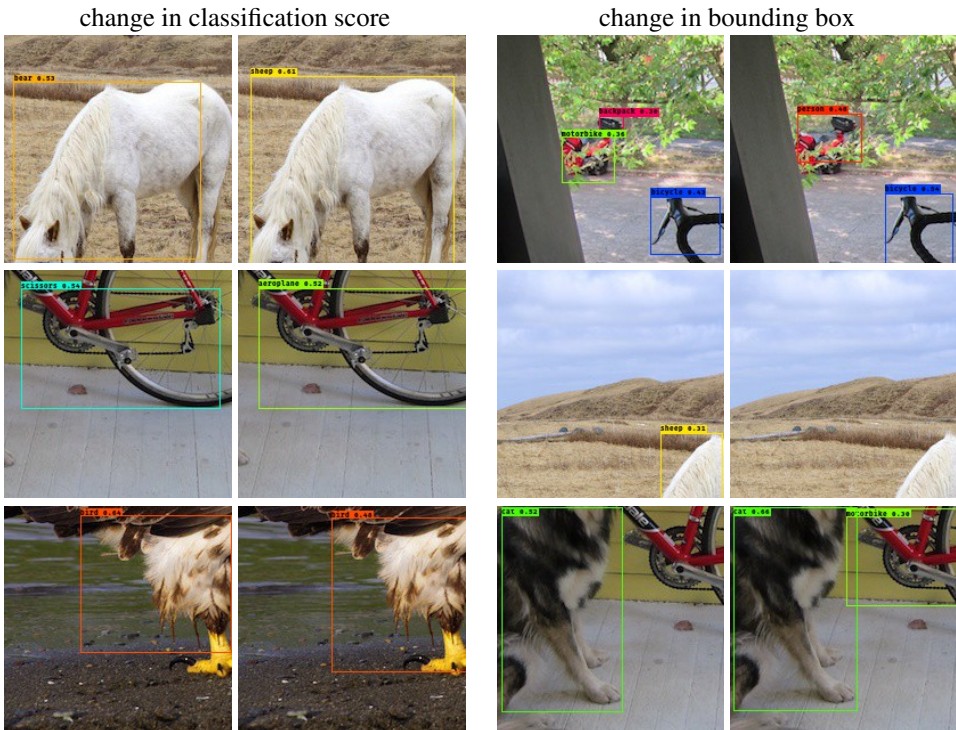

Figure A.12: *FRIs for the CNN-based "YOLO"(Redmon & Farhadi, 2017) object detection algorithm.* These examples were obtained by applying the YOLO algorithm on two adjacent windows of size $200^2$ pixels created by 1 pixel shift in the rows dimension. The results demonstrate how detection algorithms are fragile too: the output bounding boxes and their corresponding label scores are dramatically different for these two cropped regions.

