# OpenReview forum: "Minimal Images in Deep Neural Networks: Fragile Object Recognition in Natural Images"
_ICLR.cc/2019/Conference_

### Official Review · AnonReviewer1 · 2018-11-01
**ok paper; an important question on upscaling image crops**

**Rating:** 6
**Confidence:** 4

**Review:**

Thanks the authors for an interesting work!
The paper studies the differences between human and DNN vision via means of minimal images (i.e. smallest image crops that can be correctly classified).

There are a few notable take-away messages:
1. DNNs are not invariant to even tiny (1-2 px) translations of small image crops.
    - It would be more insightful if authors added comparison between DNN sensitivity to tiny translations of small image crops vs. full-size images (i.e. translation-based adversarial examples https://openreview.net/forum?id=BJfvknCqFQ ).
2. The smaller the image crops, the more sensitive DNNs become (here, more FRIs)
3. DNNs and human vision misclassify the image crops differently: (1) DNNs have almost twice more FRI(s) and (2) FRIs of human and DNNs differ in location.

Questions:

- "After extracting the region from the image, the region is resized to be of the size required by the network."
Would upscaling say a small 28x28 crop into 224x224 image here naturally negatively impact the DNN predictive performance?
That is, because typically image classifiers are trained on one (or a few) fixed resolution(s) of images.

One hypothesis here is that fragile recognition may be because the test image resolution does not match the training image resolution.
Human on the other hands, have been trained on images of variable resolutions.

An alternative to upscaling here is to zero-pad the crop region. Can you help us understand your choice of upscaling here?

+ Originality
The originality is limited as it is a close extenstion work of Ullman et al. 2016

+ Clarity
The paper is well written and presented.

+ Significance
This work extends our understanding of the differences between DNNs and human vision.
However, given what we learned from the adversarial example research area, the contribution of this work is low because results might not be too surprising.

---

> ### Author Response · Authors · 2018-11-26
> **rebuttal**
>
> We thank the reviewer for her/his valuable comments and questions, which we address below.
>
> Reviewer: “It would be more insightful if authors added comparison between DNN sensitivity to tiny translations of small image crops vs. full-size images”
> Authors: We are unsure we have understood correctly this comment because in Section 4 of the original submission (“Fragile recognition is not lack of object location invariance”) we have provided the comparison between DNN sensitivity to tiny translations of small image crops vs. full-size images.
>
> Reviewer: “Would upscaling say a small 28x28 crop into 224x224 image here naturally negatively impact the DNN predictive performance? [...] One hypothesis here is that fragile recognition may be because the test image resolution does not match the training image resolution.
> Authors: The results presented in the paper reject this hypothesis:
> 1) Matching the training image resolution does not solve FRIs. Fig. 6a shows that augmenting training data with small crops does not solve FRIs.
> 2) There are FRIs at any crop size. Fig.5 shows that about 1 out of 50 crops of 200x200 pixels (P=0.8) are FRIs.
>
> Reviewer: “An alternative to upscaling here is to zero-pad the crop region. Can you help us understand your choice of upscaling here?”
> Authors: Note that FRIs can also be constructed by zero-padding. We added an experiment to clarify this in Fig. A.6. We tested FRIs generated by zero-padding (instead of upscaling) in DNNs with different pooling region sizes. The results show that FRIs arise in zero-padding in the same way as upscaling, and larger pooling does not significantly reduce the amount of FRIs the way it reduces lack of invariance. This result further strengthens the conclusion that FRIs are a phenomenon independent of the lack of invariance reported in previous works. Upscaling was chosen for the study instead of zero-padding because upscaling leads to natural images while zero-padding does not. FRIs can be produced by just moving the camera a bit closer or a bit to the side. Studying the failure modes of DNNs in natural images is crucial, as these failures happen without the need of an attacker handcrafting images by adding zero-padding or synthetic perturbations.
>
> Reviewer: “The originality is limited as it is a close extenstion work of Ullman et al. 2016”
> Ullman et al. analyses minimal images for human vision, while we analyze minimal images for DNNs (more generally denoted as FRIs). We have shown for the first time that: 1) minimal images are a common phenomenon among human vision and DNNs, 2) DNNs are more severely affected by minimal images than humans, and 3) minimal images are a new failure mode of DNNs that arise in natural images and are independent of phenomena found in previous works.
>
>
> Reviewer: “Given what we learned from the adversarial example research area, the contribution of this work is low because results might not be too surprising.”
> Authors: Note that we have shown a new type of adversarial example that arises without the need of artificial perturbations. As shown in Section 4, previous works on adversarial examples without artificial perturbations use zero-padding and can be alleviated with architectures with large pooling regions, while FRIs are entirely natural images and cannot be similarly alleviated.

---

> > ### Comment · AnonReviewer1 · 2018-12-02
> > **differences vs. Ullman et al 2018**
> >
> > Thanks the authors for your comments!
> >
> > I agree with some comments and have a question regarding the difference between this work and Ullman et al.
> >
> > > Ullman et al. analyses minimal images for human vision, while we analyze minimal images for DNNs
> >
> > Isn't it that Ullman et al. already compared simulations vs human vision, and showed that DNNs do not recognize FRIs at the human level?

---

> > > ### Author Response · Authors · 2018-12-02
> > > **differences vs. Ullman et al 2018**
> > >
> > > Note that evaluating human FRIs on DNNs (Ullman et al.) is quite different from extracting FRIs from DNNs (our paper). These two experiments investigate different things:
> > >
> > > *evaluating human FRIs on DNNs (Ullman et al.): Are DNNs affected by human FRIs? The answer was no.
> > > *extracting FRIs on DNNs (our paper): do DNNs have their own set of FRIs (different from humans FRIs)? The answer is yes. We showed for the first time that DNNs also have FRIs.
> > >
> > > So, Ullman et al. asks about the transferability of human FRIs to DNNs, while we ask if DNNs have their own FRIs.
> > >
> > > Quoting from our paper in the introduction:
> > > "Ullman et al. (2016) show that DNNs are unable to recognize human minimal images, and the DNN
> > > drop in accuracy for these minimal images is gradual rather than sharp. This begs the question of
> > > whether the sharp drop in accuracy for minimal images is a phenomenon exclusive to human vision,
> > > or there exist distinct but analogous images that produce a sharp drop in DNN accuracy."

---

### Official Review · AnonReviewer2 · 2018-11-05
**Interesting progress in measuring the fragility of deep neural network based recognition.**

**Rating:** 7
**Confidence:** 4

**Review:**

This paper is a more thorough follow-up to e a previous work by Ullman et al that was comparing minimally recognizable patches by humans compared to deep neural network. This paper exhibits that a wide range of architectures features the same fragility and that these effects can combated by better training methodology and different pooling architectures. Still even with those changes deep CNNs still posses more fragile behavior than human vision. One of my criticism is that human vision is kind of different: it makes multiple passes over the same images at multiple scales, so this might contribute significantly to these differences. Still this paper makes a lot of interesting observations and analyses and represents a first methodological study of this phenomenon.

A novelty of this work is that it is the first paper that methodologically analyses FRIs for DNNs a reasearch area which might shed new light on the understanding of how vision systems work and the source of misrecognitions and the limitations of recognition systems.

In light of the changes of the paper and the clarification on the novelty aspect of this research, I suggest this paper to be accepted as it constitutes novel research in understanding how DNNs recognize image content and its similarities and differences to human vision.

---

> ### Author Response · Authors · 2018-11-26
> **rebuttal**
>
> We thank the reviewer for her/his valuable comments and questions, which we address below.
>
> Reviewer: “human vision is kind of different: it makes multiple passes over the same images at multiple scales, so this might contribute significantly to these differences.”
> Authors: DNNs and human vision are different and investigating these differences will help developing better DNNs and understanding human vision. In the paper, we have shown that minimal images are a common phenomenon among DNNs and humans, which opens a new line of research for studying the commonalities and differences between DNNs and humans.  To illustrate how to proceed in this line of research, we added an experiment to show the effect of multiscale and the eccentricity dependence of human vision in FRIs. Fig. A.4 shows the FRIs for a scale invariant architecture that processes multiple scales in parallel and is eccentricity dependent (Chen et al. 2017), trained in CIFAR-10. We can see that this architecture alleviates FRIs compared to the architectures we previously tested, but there is still much to do to completely close the gap between DNNs and humans.

---

### Official Review · AnonReviewer3 · 2018-11-09
**Interesting paper but requires additional experiments**

**Rating:** 7
**Confidence:** 4

**Review:**

Ullman et al. showed that slight changes in location or size of visible regions in minimal recognizable images can significantly impair human ability to recognize objects. This paper is a  follow-up of Ullman et al. paper, with focus on sensitivity of DNNs to certain regions in images. In other words, slight change of such regions’ size or location in the image can significantly affect DNN ability in recognizing them, even-though these changes are not noticeable for humans.


Comments and questions:

This paper provides in-depth study of fragile recognition in DNNs.

- Visualizing activations of different layers of DNN for Loose shift/shrink FRIs can potentially provide more details on why the final output of DNN is significantly different for two visually similar images.

- Naively augmenting training data with crops of small FRI sizes can potentially harm and confuse DNN in classifying training samples as many small patches in training images are background and they don't contain target object. It is interesting to see the sensitivity of DNNs that are trained for the task of object detection to FRIs, like sensitivity of R-CNN to FRIs. In this case augmenting training data with crops of small FRI sizes can be properly done since ground-truth bounding boxes can determine which region is foreground and which region is background.

---

> ### Author Response · Authors · 2018-11-26
> **rebuttal**
>
> We thank the reviewer for her/his valuable comments and questions, which we address below.
>
> Reviewer: “Visualizing activations of different layers of DNN for Loose shift/shrink FRIs can potentially provide more details on why the final output of DNN is significantly different for two visually similar images.”
> Authors: We have added visualizations in Figure A.11. As in previous works on adversarial examples, these visualizations do not clarify much beyond that fact that differences are small for the first layers, then are magnified at the later layers.
>
> Reviewer: “Naively augmenting training data with crops of small FRI sizes can potentially harm and confuse DNN in classifying training samples as many small patches in training images are background and they don't contain target object.”
> Authors: To verify that adding crops in the training set does not harm the accuracy, we added Fig. A.2 in the paper, which shows that the accuracy of the network always improves when adding the crops in the training set. This may be because in CIFAR the crops are always on the object, as the object is centered and occupies the whole CIFAR image. For ImageNet, the tested networks add crops in the training set from the interior of the annotated bounding-box.
>
> Reviewer: “It is interesting to see the sensitivity of DNNs that are trained for the task of object detection to FRIs, like sensitivity of R-CNN to FRIs.”
> Authors: We added Figure A.12 in the paper to show qualitative examples of FRIs for the YOLO object detector. This result illustrates that object detectors also suffer from FRIs, as they are based on DNNs. Quantifying how much the accuracy of the detectors is due to FRIs is an interesting follow up of our paper.

---

### Meta-Review · Area_Chair1 · 2018-12-10

**Confidence:** 4
**Recommendation:** Accept (Poster)

**Metareview:**

This paper characterizes a particular kind of fragility in the image classification ability of deep networks: minimal image regions which are classified correctly, but for which neighboring regions shifted by one row or column of pixels are classified incorrectly. Comparisons are made to human vision. All three reviewers recommend acceptance. AnonReviewer1 places the paper marginally above threshold, due to limited originality over Ullman et al. 2016, and concerns about overall significance.